# A Systematic Review on the Therapeutic Potentiality of PD-L1-Inhibiting MicroRNAs for Triple-Negative Breast Cancer: Toward Single-Cell Sequencing-Guided Biomimetic Delivery

**DOI:** 10.3390/genes12081206

**Published:** 2021-08-04

**Authors:** Mahdi Abdoli Shadbad, Sahar Safaei, Oronzo Brunetti, Afshin Derakhshani, Parisa Lotfinejad, Ahad Mokhtarzadeh, Nima Hemmat, Vito Racanelli, Antonio Giovanni Solimando, Antonella Argentiero, Nicola Silvestris, Behzad Baradaran

**Affiliations:** 1Research Center for Evidence-Based Medicine, Faculty of Medicine, Tabriz University of Medical Sciences, Tabriz 5166614766, Iran; abdoli.med99@gmail.com (M.A.S.); p.lotfinezhad@gmail.com (P.L.); 2Immunology Research Center, Tabriz University of Medical Sciences, Tabriz 5165665811, Iran; shar.safaee@gmail.com (S.S.); a.derakhshani@oncologico.bari.it (A.D.); nima.hemmat1995@gmail.com (N.H.); ahad.mokhtarzadeh@gmail.com (A.M.); 3Student Research Committee, Tabriz University of Medical Sciences, Tabriz 5165665811, Iran; 4Medical Oncology Unit, IRCCS Istituto Tumori Giovanni Paolo II, 70124 Bari, Italy; dr.oronzo.brunetti@tiscali.it (O.B.); antoniogiovannisolimando@gmail.com (A.G.S.); argentieroantonella@gmail.com (A.A.); 5Laboratory of Experimental Pharmacology, IRCCS Istituto Tumori Giovanni Paolo II, 70124 Bari, Italy; 6Department of Biomedical Sciences and Human Oncology, Unit of Internal Medicine and Clinical Oncology, University of Bari “Aldo Moro”, 70124 Bari, Italy; vito.racanelli1@uniba.it; 7Department of Immunology, Faculty of Medicine, Tabriz University of Medical Sciences, Tabriz 5166614766, Iran

**Keywords:** programmed death-ligand 1, PD-L1, triple-negative breast cancer, TNBC, microRNA, microRNA-based gene-therapy, biomimetic carriers, leukosomes, single-cell sequencing, personalized medicine

## Abstract

The programmed death-ligand 1 (PD-L1)/programmed cell death protein 1 (PD-1) is a well-established inhibitory immune checkpoint axis in triple-negative breast cancer (TNBC). Growing evidence indicates that tumoral PD-L1 can lead to TNBC development. Although conventional immune checkpoint inhibitors have improved TNBC patients’ prognosis, their effect is mainly focused on improving anti-tumoral immune responses without substantially regulating oncogenic signaling pathways in tumoral cells. Moreover, the conventional immune checkpoint inhibitors cannot impede the de novo expression of oncoproteins, like PD-L1, in tumoral cells. Accumulating evidence has indicated that the restoration of specific microRNAs (miRs) can downregulate tumoral PD-L1 and inhibit TNBC development. Since miRs can target multiple mRNAs, miR-based gene therapy can be an appealing approach to inhibit the de novo expression of oncoproteins, like PD-L1, restore anti-tumoral immune responses, and regulate various intracellular singling pathways in TNBC. Therefore, we conducted the current systematic review based on the preferred reporting items for systematic reviews and meta-analyses (PRISMA) to provide a comprehensive and unbiased synthesis of currently available evidence regarding the effect of PD-L1-inhibiting miRs restoration on TNBC development and tumor microenvironment. For this purpose, we systematically searched the Cochrane Library, Embase, Scopus, PubMed, ProQuest, Web of Science, Ovid, and IranDoc databases to obtain the relevant peer-reviewed studies published before 25 May 2021. Based on the current evidence, the restoration of miR-424-5p, miR-138-5p, miR-570-3p, miR-200c-3p, miR-383-5p, miR-34a-5p, miR-3609, miR-195-5p, and miR-497-5p can inhibit tumoral PD-L1 expression, transform immunosuppressive tumor microenvironment into the pro-inflammatory tumor microenvironment, inhibit tumor proliferation, suppress tumor migration, enhance chemosensitivity of tumoral cells, stimulate tumor apoptosis, arrest cell cycle, repress the clonogenicity of tumoral cells, and regulate various oncogenic signaling pathways in TNBC cells. Concerning the biocompatibility of biomimetic carriers and the valuable insights provided by the single-cell sequencing technologies, single-cell sequencing-guided biomimetic delivery of these PD-L1-inhibiting miRs can decrease the toxicity of traditional approaches, increase the specificity of miR-delivery, enhance the efficacy of miR delivery, and provide the affected patients with personalized cancer therapy.

## 1. Introduction

Breast cancer is among the frequently diagnosed malignancy among females. Yersal et al. have classified breast cancers into 5 groups, i.e., luminal-A, luminal-B, human epidermal growth factor receptor 2 (HER2)-positive, basal-like breast, and normal breast-like. Luminal-A is characterized by low proliferation-related genes expression and a high level of estrogen receptor (ER) expression and carries a relatively good prognosis. In contrast to luminal-A, luminal-B is characterized by increased expression of proliferation-related genes and carries a relatively poor prognosis. HER2-positive breast cancers are aggressive tumors characterized by increased HER2 expression and other genes related to this signaling pathway. Basal-like breast cancer is characterized by aggressive nature, high histological grade, high mitotic index, increased expression of myoepithelial markers, and no expression of ER, progesterone receptor (PR), and HER2. Clinically, they are regarded as TNBC; however, they are not completely synonymous. The normal breast-like group is a poorly defined group that usually does not respond to neoadjuvant chemotherapy. Since they do not express ER, PR, HER2, they might be regarded as triple-negative. However, their genetic profile is not consistent with basal-like breast cancers [1]. TNBC, which accounts for approximately 15% of breast cancer cases, does not overexpress the conventional receptors of breast cancer, i.e., PR, HER2, and ER [2]. Besides the lack of targets for target therapy, the highly aggressive nature of TNBC and the poor prognosis of TNBC patients can justify the need to develop novel therapeutic approaches to treat TNBC patients [3]. 

Inhibitory immune checkpoints are implicated in maintaining the immunosuppressive tumor microenvironment. The PD-L1/PD-1 axis is one of the critical and well-established inhibitory immune checkpoint axes that is highly implicated in immune evasion of tumoral cells. This axis can be established between tumoral cells and immune cells [4]. A recent meta-analysis has shown a strong association between tumor-infiltrating T-cells and PD-L1 in TNBC patients [5]. Based on a study, PD-L1 is overexpressed on 85% of TNBC, and its overexpression is associated with inferior overall survival and disease-free survival in patients with TNBC [6]. Also, it has been reported that TNBC cells express PD-L1 more than other breast cancer subtypes [7]. It has been shown that PD-L1 silencing can substantially decrease tumor viability, suppress tumor clonogenicity, arrest the cell cycle, stimulate apoptosis, inhibit tumor migration, upregulate pro-inflammatory cytokines, and downregulate anti-inflammatory cytokine a co-culture system with T-cells and TNBC cells [8]. Therefore, tumoral PD-L1 can be a valuable target for target therapy of TNBC cancer. 

miRs are noncoding RNAs of 21–25 nucleotides in length, which their dysregulated forms have been implicated in cancer development via post-transcriptional modifications of oncoproteins and tumor suppressor genes [9,10,11]. Recent studies have identified that specific miRs can downregulate tumoral PD-L1 expression and inhibit tumor development in TNBC [12,13]. Besides transforming the immunosuppressive tumor microenvironment into a pro-inflammatory one, the restoration of these PD-L1-inhibiting miRs can regulate various oncogenic signaling pathways, inhibit the de novo expression of oncoproteins, like PD-L1, in TNBC cells. 

Herein, the current systematic review aims to summarize the recent findings regarding the effect of these PD-L1-inhibiting miRs on TNBC development and anti-tumoral immune responses. Besides, this study intends to briefly discuss the recent advances in biomimetic carriers as safe and biocompatible vehicles for miR delivery to tumoral cells. Furthermore, we propose a novel strategy to increase the specificity of these carriers to increase their specificity for delivering these PD-L1-inhibiting miRs. The results of the current systematic review might provide a blueprint for future studies to bring miR-based gene therapy for treating TNBC. 

## 2. Methods

This systematic review was conducted according to the PRISMA statements [14].

### 2.1. Search Strategy

The Cochrane Library, Embase, Scopus, PubMed, ProQuest, Web of Science, Ovid, and IranDoc were systematically searched to obtain peer-reviewed records published before 25 May 2021 without any restrictions in language and country. For this systematic search, we used the following keywords: (“BT20” OR “BT-549” OR “Cal51” OR “HCC 38” OR “HCC 1143” OR “HCC 1187” OR “HCC 1395” OR “HCC 1599” OR “HCC 1739” OR “HCC 1806” OR “HCC 1937” OR “HCC 2157” OR “HCC 2185” OR “HCC 3153” OR “HS578T” OR “MDA-MB-157” OR “MDA-MB-231” OR “MDA-MB-435” OR “MDA-MB-436” OR “MDA-MB-468” OR “SUM52PE” OR “SUM102PT” OR “SUM149PT” OR “Sum159PT” OR “SUM185PE” OR “SUM229PE” OR “SUM1315M02” OR “triple” OR “triple negative breast cancer” OR “triple-negative” OR “triple negative” OR “triple-negative breast cancer” OR “TNBC” OR “HCC70” OR “DU4475” OR “Hs 578T” OR “MDA-MB-453” OR “HCC1599” OR “HCC1937” OR “HCC1143” OR “HCC38” OR “HCC1806” OR “HCC1187” OR “HCC1395” OR “CRL-2331” OR “CRL-2336” OR “CRL-2321” OR “HTB-132” OR “CRL-2314” OR “CRL-2315” OR “CRL-2335” OR “CRL-2322” OR “HTB-123” OR “HTB-122” OR “HTB-126” OR “HTB-26” OR “HTB-130” OR “HTB-24” OR “HTB-131” OR “HTB-19” OR “CRL-2324” OR “triple negative breast neoplasms”) and (“CD274” OR “programmed death-ligand 1” OR “PD-L1” OR “PDL1” OR “B7-H1” OR “PDCD1LG1” OR “B7 H1” OR “B7H1” OR “HPD-L1” OR “cluster of differentiation 274” OR “PD L1” OR “CD 274” OR “cluster of differentiation274” OR “B7 homolog 1” OR “PDCD1 ligand 1” OR “PDCD1L1” OR “B7-H1 antigen” OR “programmed death 1 ligand 1”) and (“miRNA-” OR “microRNA-” OR “microRNAs” OR “miR-“ OR “micro RNA” OR “microRNA” OR “micro RNA-” OR “miRNA” OR “miR” OR “miRNA-” OR “miRNA”). We extracted the names of TNBC cell lines from the study conducted by Chavez et al. [15]. We also incorporated Emtree and MeSH terms to increase the sensitivity of our systematic review. 

### 2.2. Study Selection

Following the systematic search, the obtained records were selected in two phases. In phase I, two authors (M.A.S. and S.S.) independently screened the studies based on their titles and abstracts. In phase II, those authors independently reviewed the full texts of the remaining studies, along with their supplementary data, to consider them to be included in the current systematic review. 

### 2.3. Eligibility Criteria

For the current systematic review, we included the studies that met the following inclusion criteria: (1) peer-reviewed original papers published in English or Persain, (2) studies investigating miR restoration on PD-L1 expression in TNBC cells. Based on the following criteria, these studies were excluded from the current systematic review: (1) studied that failed to fulfill the above-mentioned criteria, (2) studies that investigated the effect of a pro-PD-L1 miR, (3) studies that investigated the cross-talk between studied miR restoration and PD-L1 expression in other cells residing in the tumor microenvironment, rather TNBC cells, (4) studies that their data were solely based on bioinformatics, and (5) studies that investigated the effect of miR restoration on PD-L1 in non-human TNBC cells.

### 2.4. Data Extraction

The following data were extracted from the included studies: the first author, the publication year, the studied miR, the effect of studied miR on TNBC cell/the tumor microenvironment of TNBC, and studied TNBC cell line.

### 2.5. Evaluating the Potential Bias among the Included Studies

To increase the transparency of the obtained results and highlight the potential risk of bias among the included studies, two quality assessment tools were used. For in vivo studies that investigated the restoration of PD-L1-inhibiting miR in animal models, we used the SYRCLE’s RoB checklist [16]. For in vitro studies, we used a previously applied checklist for assessing the potential risk of bias among in vitro studies [17].

### 2.6. In Silico Investigation

Furthermore, the data from the WikiPathways was obtained to investigate the effect of these PD-L1-inhibiting miRs on the biological pathways. As previously described, the data from the miRPathDB v2.0 (https://mpd.bioinf.uni-sb.de/ (accessed on 12 June 2021)) were analyzed with adjusting on the strong experimental evidence and a minimum of two significant miRs per pathway [11].

## 3. Results

### 3.1. Selected Studies

Our systematic search has retrieved 178 peer-reviewed studies from Embase (*n* = 57), Scopus (*n* = 39), Ovid (*n* = 28), Web of Science (*n* = 27), PubMed (*n* = 23), IranDoc (*n* = 3), ProQuest (*n* = 1), and the Cochrane Library (*n* = 0). After removing duplicated records, 62 records remained. In phase I, we excluded 38 records based on their titles and abstracts. In phase II, we excluded 12 studies based on reviewing their full texts and their supplementary data. The exclusion reasons for these 12 studies are demonstrated in Table 1. Two authors (M.A.S. and S.S.) conducted the study selection independently. The flowchart of literature identification and inclusion is demonstrated in Figure 1. 

### 3.2. Study Characteristic

The included studies were published between 2017 and 2021. We have found that the restoration of miR-424-5p, miR-138-5p, miR-570-3p, miR-200c-3p, miR-383-5p, miR-34a-5p, miR-3609, miR-195-5p, and miR-497-5p can substantially downregulate PD-L1 expression in TNBC cells and inhibit TNBC development. The MDA-MB-231 cell line was the most frequently studied cell line in the included studies. The summary of the included studies is present in Table 2.

### 3.3. The Evaluation of Potential Bias in the Included Studies

Our results have indicated that the concentration of studied miR is the main area of bias in the included in vitro studies; however, the overall risk of bias has not been considerable enough to endanger the obtained results from those in vitro studies (Table 3). Besides, our results have shown that random housing of animal models and blinding (detection bias) are the main areas of bias in the included in vivo studies (Table 4). Nevertheless, the overall risk of bias has not been remarkable enough to endanger the obtained results from those in vivo studies.

### 3.4. In Silico Investigation

Our obtained results have indicated that miR-383-5p, miR-195-5p, and miR-497-5p are significantly enriched for apoptosis (Figure 2). Besides, miR-424-5p, miR-34a-5p, miR-195-5p, and miR-497-5p are significantly enriched for G1 to S cell cycle control category (Figure 2). Moreover, miR-424-5p, miR-383-5p, miR-200c-3p, miR-34a-5p, miR-195-5p, and miR-497-5p are significantly enriched for the phosphatidylinositol-3-kinase and protein kinase B (PI3K/Akt) pathway (Figure 2).

## 4. Discussion

Since the PD-L1/PD-1 immune checkpoint axis can pave the way for the immune evasion of tumoral cells, multiple antibodies targeting this axis have been investigated in clinical trials [4]. Indeed, this axis can partially shield tumoral cells from anti-tumoral immune responses. Besides, growing evidence indicates that PD-L1 can mediate oncogenic signaling pathways, leading to tumor proliferation and tumor migration [8,39,40]. Following the satisfactory results of immune checkpoint inhibitors on stimulating anti-tumoral immune responses, the FDA has approved pembrolizumab, a monoclonal antibody targeting PD-1, for locally recurrent unresectable/metastatic TNBC [39]. However, the current form of inhibitory immune checkpoint blockade cannot suppress the de novo expression of immune checkpoints in tumoral cells. Besides, these monoclonal antibodies cannot substantially regulate oncogenic signaling pathways in TNBC cells. Table 5 summarizes the recent trend in TNBC immunotherapy.

In this regard, recent studies have identified a group of miRs that their restoration can transform immunosuppressive tumor microenvironment into pro-inflammatory one via inhibiting tumoral PD-L1 expression in TNBC cells and regulate oncogenic signaling pathways in TNBC cells. Herein, we shed light on these PD-L1-inhibiting miRs and elucidate their therapeutic potentiality in inhibiting TNBC development. Besides, we briefly discuss the recent advances in biomimetic carriers for the delivery of these PD-L1-inhibiting miRs. In this approach, the current challenges in gene therapy of tumoral cells, i.e., the toxicity of traditional nanoparticles and biocompatibility of miR delivery, are addressed. Also, we depict the future perspectives to optimize the delivery of PD-L1-inhibiting miRs in terms of specificity and targeted therapies. Finally, we mention some critical points in the application of miR-based gene therapy for cancer treatment.

### 4.1. miRs That Can Downregulate Tumoral PD-L1 Expression in TNBC Cells

#### 4.1.1. miR-424-5p

Dastmalchi et al. have shown that miR-424-5p restoration can suppress the PI3K/Akt pathway and inhibit angiogenesis in MDA-MB-231 cells. Moreover, its overexpression can arrest the cell cycle, stimulate apoptosis, and inhibit the stemness feature of TNBC cells. In a co-culture system with T-cells, miR-424-5p restoration has decreased the protein expression of PD-L1, upregulated pro-inflammatory cytokines, and downregulated interleukin (IL)-10 expression [34]. It has been shown that miR-424-5p can enhance the chemosensitivity of MDA-MB-231 cells to taxol, and the combination therapy with miR-424-5p and taxol has shown superiority in terms of stimulating apoptosis, arresting cell-cycle, suppressing the clonogenicity of tumoral cells, and inhibiting the PI3K/Akt signaling pathway over monotherapy with taxol [30]. Zhou et al. have also shown that miR-424-5p can inhibit PD-L1 mRNA in MDA-MB-231 cells and stimulate pro-inflammatory secretion cytokines, i.e., interferon (IFN)-γ, tumor necrosis factor (TNF)-α, and IL-6, and downregulate IL-10 in the tumor microenvironment. Indeed, the vesicle-mediated transportation of miR-424-5p has promoted T-cell-mediated apoptosis and decreased tumor size in animals bearing TNBC. Their bioinformatic results from the METABRIC have demonstrated that miR-424-5p expression can be associated with improved overall survival in breast cancer patients (HR = 0.77, 95% CI: 0.63−0.94, and *p*-value = 0.0091) [12]. Wang et al. have demonstrated that miR-424-5p is substantially downregulated in TNBC tissues and MDA-MB-468 cells compared to adjacent normal tissues and MCF-10A. Besides, miR-424-5p has substantially decreased the expression of doublecortin-like kinase 1 (DCLK1) and suppressed the migration and invasion of MDA-MB-468 cells [40]. Besides, Xie et al. have reported that miR-424-5p can target cyclin-dependent kinase 1 (CDK1) in MDA-MB-231, decrease the clonogenicity of tumoral cells, and arrest the cell cycle at the G2/M phase [41]. Consistent with these, our obtained results have indicated that miR-424-5p is significantly enriched for the G1 to S cell cycle control category and the PI3K/Akt signaling pathway (Figure 2).

#### 4.1.2. miR-138-5p

Rasoolnezhad et al. have shown that miR-138-5p can substantially downregulate tumoral PD-L1, increase INF-γ/TNF-α expression, and decrease IL-10 expression in a co-culture system with T-cells. The restoration of miR-138-5p has been associated with downregulated matrix metalloproteinase (MMP)-2, MMP-9, and vimentin and upregulated E-cadherin expression in MDA-MB-231 cells [32]. Zhao et al. have demonstrated that the expression of miR-138-5p is substantially downregulated in MDA-MB-231 cells compared to MCF-10A. Besides, miR-138-5p restoration has considerably decreased the migration and invasion ability of MDA-MB-231 cells [42]. Also, Li et al. have demonstrated that miR-138-5p can inhibit the enhancer of zeste homolog 2 (EZH2), and its restoration can arrest the cell cycle at the G1 phase and suppress the migration and proliferation of TNBC cells [43]. Consistent with these, Dengfeng et al. have demonstrated that miR-138-5p can considerably suppress the proliferation, migration, invasion, clonogenicity of TNBC cells, and its restoration can downregulate MMP-2, MMP-9, CDK2, CDK4, CDK6, cyclin D1, and cyclin E in MDA-MB-231 cells [44].

#### 4.1.3. miR-570-3p

In MDA-MB-231 and MDA-MB-468 cells, miR-570-3p can inhibit PD-L1 expression, tumor proliferation, and migration. Wang et al. have suggested that miR-570-3p can suppress TNBC via suppressing the PI3K/Akt/mammalian target of rapamycin (mTOR) signaling pathway. Besides, the expression level of miR-570-3p is substantially decreased in TNBC tissues compared to adjacent normal tissues [13].

#### 4.1.4. miR-200c-3p

Peng et al. have shown that the inhibition of miR-200c-3p has upregulated tumoral PD-L1 in MDA-MB-231 cells [33]. Ren et al. have shown that miR-200c-3p expression is considerably downregulated in MDA-MB-231 cells and TNBC tissues compared to MCF-10A and adjacent normal tissues, respectively. Besides, miR-200c-3p has decreased the colony numbers, stimulated apoptosis of TNBC cells, and decreased tumor growth in vivo [45]. Besides transforming the immunosuppressive tumor microenvironment into a pro-inflammatory one via inhibiting PD-L1, Zhang et al. have demonstrated that miR-200-3p can suppress ANLN in TNBC cells. ANLN has been associated with chemoresistance in breast cancer, and its silencing can stimulate apoptosis, decrease cell viability of tumoral cells, and increase cleaved-caspase-3 protein level in TNBC cells [46]. Liu et al. have reported that the transfection of miR-200-3p-inhibitor has substantially increased the invasion and migration of MDA-MB-231 cells [46]. Chang et al. have demonstrated that miR-200-3p can target high mobility group box protein 1 (HMGB1) and suppress the migration and invasion of TNBC cells [47]. Consistent with these, Berber et al. have reported that the expression of miR-200-3p is substantially decreased in metastatic TNBC tumors compared to non-metastatic tumors [48]. Our in silico results have shown that miR-200-3p is significantly enriched for the PI3K/Akt singling pathway.

#### 4.1.5. miR-383-5p

Azarbarzin et al. have shown that miR-383-5p can downregulate tumoral PD-L1 and inhibit the invasion, clonogenicity, and proliferation of MDA-MB-231 cells. Moreover, its restoration can stimulate apoptosis and arrest the cell cycle of tumoral cells at the sub-G1 phase. Furthermore, miR-383-5p can upregulate IL-2, TNF-α, and INF-γ and downregulate IL-10 and transforming growth factor (TGF)-β in a co-culture system with T-cells. The anti-tumoral effects of miR-383-5p can be associated with its inhibitory effect on the PI3K/Akt/mTOR signaling pathway [35]. It has been reported that the expression of miR-383-5p is substantially decreased in MDA-MB-231 cells compared to MCF-10A cells. Besides, its restoration has remarkably inhibited the migration, invasion, and proliferation of MDA-MB-231 and MDA-MB-453 cells. In breast cancer patients, its downregulation has been associated with the inferior overall survival of affected patients (HR = 2.085, 95% CI: 1.042–4.171, *p*-value = 0.038) [49]. Consistent with this, it has been shown that miR-383-5p can downregulate the expression of TWIST, signal transducer and activator of transcription (STAT)-3, vascular endothelial growth factor (VEGF), zinc finger E-box binding homeobox 1 (ZEB1), and mitogen-activated protein kinase kinase (MEK)1/2 in MDA-MB-231 cells [50]. Consistent with these, our in-silico results have indicated miR-383-5p is enriched for apoptosis and the PI3K/Akt singling pathway.

#### 4.1.6. miR-34a-5p

Zhao et al. have indicated that miR-34a-5p restoration can inhibit tumoral PD-L1 and decrease tumor proliferation and migration of MDA-MB-231 cells both in vitro and in vivo [36]. Li et al. have shown that miR-34a-5p is substantially downregulated in MDA-MB-231cells compared to MCF-10A cells [51]. Hagi et al. have shown that miR-34a-5p restoration can substantially arrest the cell cycle at the G phase, decrease tumoral cell proliferation, stimulate apoptosis, suppress the epithelial-mesenchymal transition (EMT) process, and inhibit the TGF-β singling pathway in MDA-MB-231 cells [52]. TGF-β can establish multiple immunosuppressive loops in the tumor microenvironment of breast cancer. Besides its oncogenic property in the advanced tumor grades, it can pave the way for the infiltration of myeloid-derived suppressor cells (MDSCs) into the tumor microenvironment. MDSCs have been implicated in attenuated anti-tumoral immune responses and maintaining the immunosuppressive tumor microenvironment [53,54]. Mansoori et al. have also shown that miR-34a-5p restoration can substantially decrease tumor volume in mice bearing TNBC. They have demonstrated that miR-34a-5p expression level is negatively associated with EMT score, and its restoration has substantially decreased clonogenicity and migration of MDA-MB-231 cells [55]. Huang et al. have also shown that miR-34a can downregulate PD-L1 expression and decrease the invasion and viability of tumoral cells. In animal models, miR-34a has decreased tumoral PD-L1 expression, facilitated the recruitment of CD8^+^ and CD4^+^ T-cells, and inhibited the recruitment of Tregs and macrophages [6]. Soufiani et al. have also shown that miR-34a restoration can inhibit tumoral PD-L1, decrease tumor migration, and stimulate apoptosis in MDA-MB-231 cells [31]. Our in silico results have shown that miR-34a-5p is enriched for G1 to S cell cycle control and the PI3K/Akt singling pathway.

Moreover, the delivery of miR-34a-5p via liposomes has decreased the migration, proliferation, stemness of MDA-MB-231 cells. Of interest, miR-34a-5p has substantially decreased the tumoral cells with CD44^+^CD24^−^ phenotype, which are implicated in metastasis in TNBC patients [56,57,58]. Valcourt et al. have developed poly lactic-co-glycolic acid (PLGA)-based nanoparticles fabricated with antibodies against Notch1 to deliver miR-34a-5p to TNBC cells. Their results have indicated that this complex can substantially decrease the proliferation and migration of TNBC cells. Besides increasing the specificity, inhibiting Notch1, an overexpressed factor in TNBC cells, has substantially downregulated the downstream oncogenic signals in TNBC cells [59]. Wang et al. have developed a hyaluronic acid-based biodegradable nanoparticle for the delivery of miR-34a-5p to TNBC cells. Their results have indicated that CD44, a transmembrane glycoprotein that binds to hyaluronic acid, is substantially overexpressed in MDA-MB-231 cells compared to MCF-7 cells. Besides, miR-34a delivery has remarkably downregulated the expression of Notch1, inhibit tumor migration, and decrease the tumor size and volume in mice bearing MDA-MB-231 cells [60].

#### 4.1.7. miR-3609

Li et al. have indicated that miR-3609 restoration can downregulate tumoral PD-L1 in MDA-MB-231 and MDA-MB-468 cells. Besides, the expression of miR-3609 is substantially decreased in MDA-MB-231 and MDA-MB-468 cells compared to HBL-100. In mice bearing 4T1 tumor cells, inhibiting CD8^+^ T-cells has reversed the miR-3609-mediated improved survival, indicating that miR-3609 can exert the tumor suppressor role via inhibiting tumoral PD-L1 expression and inducing the pro-inflammatory tumor microenvironment [37]. Besides, Fitzpatrick et al. have shown that miR-3609 can inhibit CDK1 in MDA-MB-231 cells [61].

#### 4.1.8. miR-195-5p

Yang et al. have shown that miR-195-5p can downregulate PD-L1 expression in MDA-MB-231 cells [38]. Luo et al. have reported that miR-195-5p can substantially arrest the cell cycle, inhibit proliferation, and decrease the migration of MDA-MB-231 cells [62]. Wang et al. have reported that miR-195-5p is substantially decreased in MDA-MB-435 and MDA-MB-231 cells compared to MCF-10A cells. Besides, miR-195-5p has remarkably inhibited tumor growth and decreased the expression of VEGF in TNBC cells [63]. Also, it has been demonstrated that miR-195-5p can inhibit the expression of KIF23, an overexpressed oncoprotein in TNBC cells implicated in proliferation, migration, and the EMT process activation, suppress the migration of tumoral cells, and decrease the number of colonies in MDA-MB-231 cells [64]. Consistent with these, our in-silico study has indicated that miR-195-5p is enriched for the PI3K/Akt signaling pathway, apoptosis, and G1 to S cell cycle control.

#### 4.1.9. miR-497-5p

In MDA-MB-231 cells, miR-497-5p restoration can inhibit tumoral PD-L1 [38]. Li et al. have shown that miR-497-5p can substantially decrease the cell viability and colony numbers of MDA-MB-468 cells. Besides, miR-497-5p has remarkably decreased the protein expression of cyclin D1 and increased the protein expression of p21 in MDA-MB-468 cells [65]. It has been reported that miR-497-5p has been considerably downregulated in TNBC tissues, MDA-MB-453, MDA-MB-468, and MDA-MB-231 cells compared to non-tumoral tissues and MCF-10A cells [66,67]. Bai et al. have demonstrated that LINC00473, an overexpressed long noncoding RNA in TNBC cells implicated in inhibiting apoptosis and increasing proliferation, migration, and clonogenicity of MDA-MB-231 cells, can target miR-497-5p in MD-MB-231 cells. Indeed, this oncogenic long-non coding RNA might be the underlying reason for the downregulated expression of miR-497-5p in TNBC cells [68]. Li et al. have demonstrated that the expression of miR-497-5p is substantially decreased in TNB patients with lymph node involvement and TNBC patients with advanced tumor stage. Besides, increased expression of miR-497-5p has been associated with improved overall survival of TNBC patients. Their in vitro results have indicated that miR-497-5p can remarkably arrest the cell cycle, stimulate apoptosis, and decrease the clonogenicity, invasion, and proliferation of MDA-MB-231 and MDA-MB-468 cells [69]. In line with these, our in-silico results have indicated that miR-497-5p is considerably enriched for apoptosis, G1 to S cell cycle control category, and the PI3K/Akt singling pathway.

### 4.2. The Delivery of miRs for Treating TNBC: Toward Effective, Biocompatible, and Safe Delivery

miR-based gene therapy is an appealing approach for treating malignancies. Although miR-based gene therapy has not been investigated in phase III clinical trials, it is a promising approach to modulate the expression of oncoproteins [70]. However, the safety, specificity, and efficacy of their delivery might be considered a daunting challenge. The ideal vehicle for miR delivery should specifically deliver cargo to tumoral cells and not induce toxicity. In this regard, polymer-based vectors and biomimetic carriers have been investigated in preclinical settings [71]. Herein, we aim to discuss the recent advances in miR-delivery via biomimetic carriers briefly. Besides, we intend to introduce recent advancements in single-cell sequencing and highlight the potentiality of combining this technology with biomimetic carriers to provide the affected patients with personalized, specific, and safe tumor-suppressive miR delivery.

#### 4.2.1. Biomimetic Carriers

The fabrication of membranes of red blood cells, white blood cells, platelets, and tumoral cells can decrease the immunogenicity and clearance of carriers. Besides, they can provide the carriers with targeted release and enhance their biocompatibility. In this regard, the biomimetic carriers derived from the leukocytes membrane have demonstrated promising results because of their inherited ligands. During inflammation, the interaction between the endothelial cells and circulating leukocytes paves the way for the infiltration of leukocytes into the inflamed site [72]. Therefore, coating the leukocyte membrane into carriers can provide us with the targeted release. Besides, the biocompatible nature of coated membranes can improve the biocompatibility of carriers and decrease toxicity and fast clearance from circulation. Since discussing all types of biomimetic carriers is out of the scope of the current study, we briefly discuss the recent advances in this field for cancer therapy.

Leukosomes are a recently introduced class of biomimetic nanovesicles that contain the specific molecules for cellular adhesion, e.g., P-selectin glycoprotein ligand-1 and lymphocyte function-associated antigen 1, among others. Recently, Molinaro et al. have developed leukosomes that can preserve the pharmaceutical properties of liposomes, and they predominantly accumulate in the tumor microenvironment in mice bearing TNBCs [73]. Martinez et al. have shown that the tumor microenvironment accumulation ability of leukosomes compared to liposomes is 16-fold. Also, leukosomes, compared to liposomes, have been predominately present in the tumoral vessels wall and lumen. However, the blockade of lymphocyte function-associated antigen 1 has substantially (around 95%) decreased the ability of leukosomes to accumulate in the tumor [74].

Neutrophils are the most abundant leukocytes in the human. Since activated neutrophils can target circulating tumor cells and tumor microenvironment, neutrophil membrane-coated carriers have been investigated for targeted delivery. Wang et al. have demonstrated that neutrophil membrane-coated carriers can escape from macrophage-mediated clearance and accumulate in the inflamed site [75]. Kang et al. have demonstrated that coating neutrophil membrane to PLGA-based nanoparticles can remarkably capture circulating tumor cells both in vitro and in vivo. Besides, these neutrophil membrane-coated carriers have been mainly accumulated in the premetastatic niche of mice bearing TNBC cells [76]. Cao et al. have reported that neutrophil membrane-coated polyethylene glycol (PEG)-PLGA-based carriers selectively accumulate in the tumor microenvironment following systematic administration. Besides, loading cytotoxic agents to this complex can substantially prolong the survival of animal models, decrease tumor size, and inhibit tumor metastasis [77]. Despite the promising results of neutrophil membrane-coated carriers in inhibiting tumor development, the inflamed sites other than the tumor microenvironment might facilitate their accumulation and decrease their specificity. Therefore, further investigations are needed to address this issue.

Unlike innate immune cells, T-cells can specifically recognize antigens. For developing T-cell-mediated immune responses, antigen-presenting cells should present antigens to naïve T-cells. Afterwards, naïve T-cells differentiate into CD4^+^ or CD8^+^ T-cells. The vast repertoire of T-cells can provide ample opportunity to develop specific T-cells for each antigen. Therefore, T-cell-membrane-coated carriers can pave the way for introducing specific carriers for miR delivery [78]. Although recent advances in single-cell sequencing have allowed us to investigate and profile tumoral cells at single-cell levels (see below), the high specificity of T-cell receptors might be an obstacle to address the vast inter-and intra-heterogeneity of tumor bulk. In this regard, metabolic glycoengineering can introduce different chemical groups to cellular glycans, which can serve as promising ligands for targeted therapy. Han et al. have intratumorally administrated bicyclo [6.1.0] nonyne (BCN) modified unnatural sugars (Ac_4_ManN-BCN) to produce ligands for their T-cell-membrane-coated carriers. Their azide-based T-cell-membrane-coated carriers have substantially accumulated in the tumor microenvironment [79]. Recently, Kang et al. have proposed another strategy to facilitate anti-tumoral immune responses via the application of T-cell-membrane-coated carriers. They have developed PLGA-based T-cell-membrane-coated carriers to stimulate anti-tumoral immune responses and deliver cytotoxic agents. Their proposed mechanisms for tumor rejection have been based on five items. First, these nanoparticles can be actively accumulated in the tumor microenvironment due to the presence of inherited lymphocyte function-associated antigen 1 on the nanoparticles and intercellular adhesion molecule-1 on the affected endothelial cells in the tumor tissue. Second, their infiltration is not influenced by the highly immunosuppressive tumor microenvironment. Third, the inherited Fas ligand of nanoparticles can activate apoptosis in tumoral cells. Fourth, they can occupy inhibitory immune checkpoint molecules and liberate the anti-tumoral immune cells from the immune checkpoint-mediated exhaustion. Besides, they can be loaded with cytotoxic factors to attenuate tumoral proliferation. Indeed, their in vivo results have indicated that these biomimetic carriers can remarkably prolong the survival of animal models [80]. Another approach to specifically target tumoral cells is developing CAR-T-cell-membrane-coated carriers. Recently, Ma et al. have extracted the membrane of CAR-T-cells that specifically target glypican-3, which is highly expressed in the studied tumor cells. Their results have indicated that delivery via CAR-T-cell-membrane-coated carriers can remarkably decrease the tumor weight compared to delivery via T-cell-membrane-coated carriers and mesoporous silica nanoparticles in animal models [81].

#### 4.2.2. Single-Cell Sequencing Technologies and Biomimetic Carriers: Future Perspectives

Single-cell sequencing has provided ample opportunities to study tumoral cells at the single-cell level. The enormous data provided by single-cell sequencing technologies have furthered our knowledge of the vast inter-and intra-heterogeneity and provided answers for the failure of conventional anti-cancer therapies. Indeed, tumor bulk is not considered cells that have unified gene mutations rather the subpopulations of tumor bulk with distinct genetic properties [82]. The heterogeneity among the tumoral cells might be a daunting challenge for identifying suitable targets for active-targeting delivery. However, single-cell sequencing can allow us to identify personalized neo-antigens for each affected patient. Ideally, these neo-antigens should be expressed in tumor cells, and the healthy cells should minimally (or ideally not) express these antigens [83]. Therefore, fabricating the pertained CAR or T-cell receptor (TCR) or peptides to the above-mentioned biocompatible carriers can ameliorate the delivery of the PD-L1-inhibiting miRs and substantially inhibit TNBC development (Figure 3). Although the recent study by Ma et al. has demonstrated promising results about the efficacy of specific CAR-T-cell-membrane-coated carriers [81], further studies are needed for the translation of this novel approach into clinical practice.

### 4.3. Some Considerations in miR-Based Therapy for Treating Cancer

Although the current systematic review has highlighted the promising results of PD-L1-inhibiting miRs restoration in treating TNBC, some points are worthy of mentioning. First, the recent advances in bioinformatic technologies have provided enormous data to predict the interactions between genes and miRs; therefore, these data can predict that specific miRs can target specific oncoproteins, like PD-L1. However, solely relying on these data, without investigating the overall effect of that miR restoration on the cell viability, migration, chemosensitivity, clonogenicity, apoptosis, etc., might be misleading. For instance, Chen et al. have reported that miR-191-5p can substantially inhibit PD-L1 expression in RKO cells, and they have not studied the overall effect of its restoration in colorectal cancer cells [84]. However, it has been reported that miR-191-5p can increase tumorigenicity and migration of colorectal cancer cells [85,86].

The second consideration should be devoted to the importance of the biological differences of -3p and -5p miRs. For instance, Yang et al. have shown that inhibiting miR-193a-5p and docetaxel treatment has remarkably decreased prostate cancer volume compared to treatment with docetaxel and control miR in mice models [87]. However, Liu et al. have demonstrated that miR-193a-3p transfection can decrease tumor proliferation, arrest the cell cycle, inhibit cell viability, and suppress clonogenicity of prostate cancer cells [88].

The third consideration is related to safe, specific, and effective miR delivery because one tumor-suppressive miR in one cancer can promote tumorigenicity in other tissues. For instance, Mansoori et al. have demonstrated that miR-142-3p can stimulate apoptosis and arrest the cell cycle at the G2/M phase in breast cancer cells [89]. However, Zhu et al. have shown that miR-142-3p can substantially inhibit apoptosis and increase the cell viability of gastric cancer cells [90]. Therefore, there is a need to develop a specific, safe, and effective miR delivery approach. 

## 5. Conclusions

The current systematic review has demonstrated that the restoration of miR-200c-3p, miR-424-5p, miR-138-5p, miR-34a-5p, miR-570-3p, miR-383-5p, miR-3609, miR-195-5p, and miR-497-5p can inhibit tumoral PD-L1 expression, transform immunosuppressive tumor microenvironment into the pro-inflammatory one, inhibit tumor proliferation, suppress tumor migration, enhance chemosensitivity of tumoral cells, stimulate tumor apoptosis, arrest cell cycle, repress the clonogenicity of tumoral cells, and regulate various oncogenic signaling pathways in TNBC cells. Concerning the safe, effective, and specific delivery of these PD-L1-inhibiting miRs to treat TNBC, the single-cell squelching-guided biomimetic-based delivery can be a promising approach in terms of its biocompatibility, specificity, safety, efficacy, and low toxicity. However, further studies are needed before the translation of this personalized medicine-based therapy into clinical practice.

## Figures and Tables

**Figure 1 genes-12-01206-f001:**
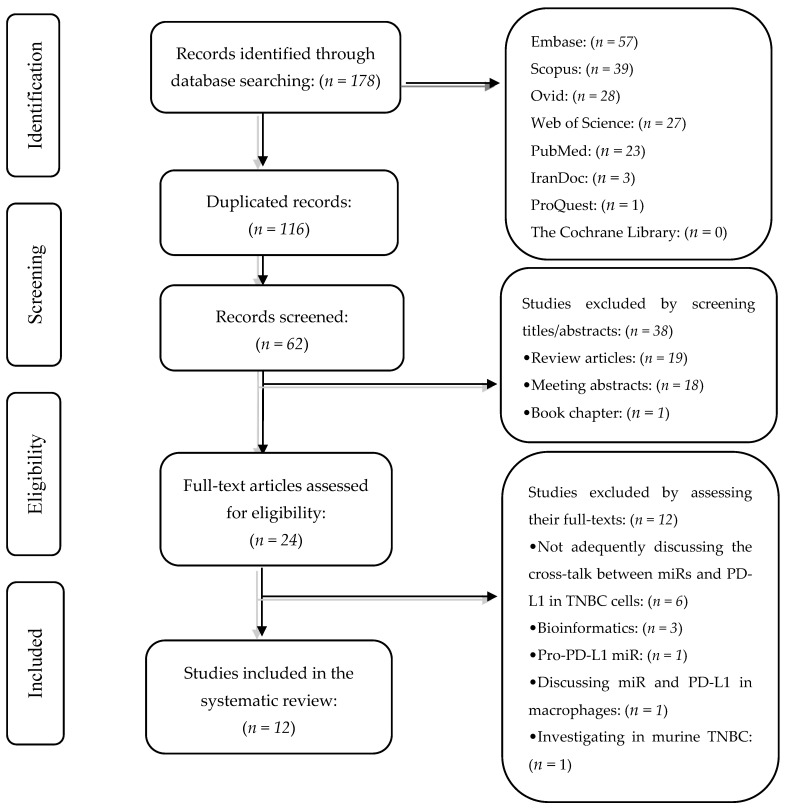
The flowchart of the study selection process.

**Figure 2 genes-12-01206-f002:**
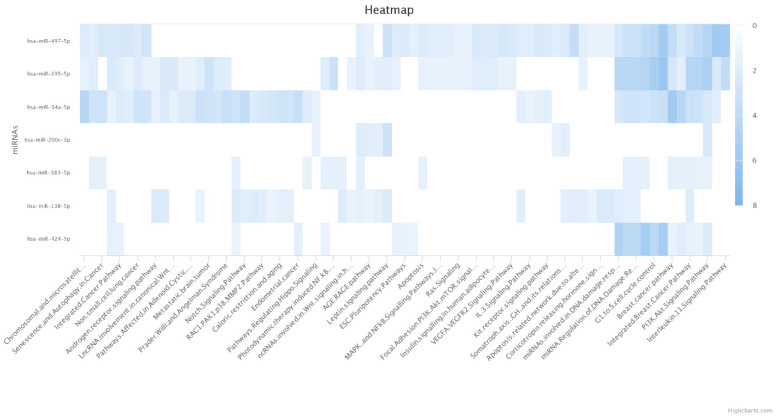
Enriched WikiPathways of PD-L1-inhibiting miRs. The darker color indicates a more significant enrichment.

**Figure 3 genes-12-01206-f003:**
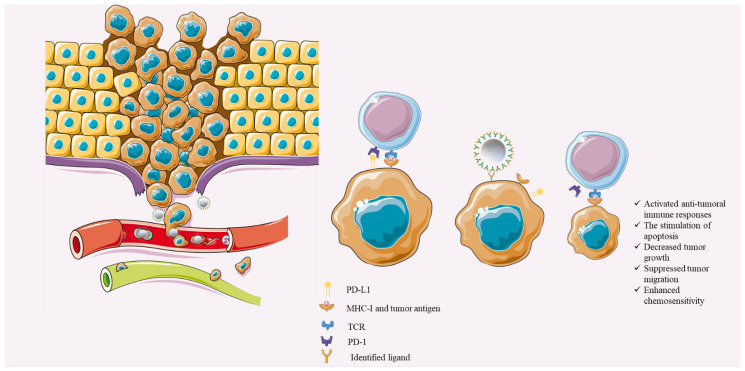
Single-cell sequencing-guided biomimetic-based miR-delivery. After identifying the tumoral antigen based on single-cell sequencing data and fabricating related receptor/CAR/TCR on the biomimetic carriers, the delivery of the PD-L1-inhibiting miRs can substantially stimulate anti-tumoral immune responses, activate apoptosis of tumoral cells, inhibit tumor proliferation, suppress tumor migration, and enhance the chemosensitivity of tumoral cells to chemotherapeutic agents. Objects are obtained from https://smart.servier.com/ (accessed on 25 July 2021).

**Table 1 genes-12-01206-t001:** The excluded studies and the reasons for their exclusion.

No.	Excluded Study in the Second Phase	The Reason for Its Exclusion
1	Peng et al. [18]	They were solely based on bioinformatics.
2	Qattan et al. [19]	
3	Liu et al. [20]	
4	Naba et al. [21]	The PD-L1-inhibiting miRs were not adequately investigated.
5	Yang et al. [22]	
6	Noman et al. [23]	
7	Zhang et al. [24]	
8	Nafea et al. [25]	
9	Youness et al. [26]	
10	Hamed et al. [27]	Pro-PD-L1 miR was studied.
11	Yao et al. [28]	The cross-talk between miR and PD-L1 in macrophages was studied.
12	Rogers, 2018 [29]	The cross-talk between miR and PD-L1 was evaluated only in murine TNBC.

Abbreviations: TNBC: triple-negative breast cancer, and PD-L1: programmed death-ligand 1.

**Table 2 genes-12-01206-t002:** The main findings of the included studies.

No.	First Author and Year	The Studied miR	The Effect of Studied miR on TNBC Cell/the Tumor Microenvironment of TNBC	The Studied Cell Line(s)
1	Zhou, 2021 [12]	miR-424-5p	This miR can inhibit tumoral PD-L1, induce a pro-inflammatory tumor microenvironment, and stimulate apoptosis in vitro. Also, its intratumoral administration can decrease tumor size in animal models.	MDA-MB-231
2	Dastmalchi, 2021 [30]	miR-424-5p	This miR can inhibit PD-L1 expression and enhance the chemosensitivity of tumoral cells to taxol. The combination therapy with miR-424-5p and taxol has shown superiority in terms of stimulating apoptosis, arresting cell-cycle, suppressing the clonogenicity of tumoral cells, and inhibiting the PI3K/Akt signaling pathway over monotherapy with taxol.	MDA-MB-231
3	Soufiani, 2021 [31]	miR-34a	This miR can substantially inhibit tumoral PD-L1 expression, decrease tumor migration, and stimulate apoptosis in tumoral cells.	MDA-MB-231
4	Rasoolnezhad, 2021 [32]	miR-138-5p	This miR can downregulate PD-L1 expression and inhibit the PI3K/Akt signaling pathway. Restoration of miR-138-5p has been associated with decreased MMP-2, MMP-9, and vimentin and increased E-cadherin expression. Besides, its restoration has stimulated apoptosis, arrested cell-cycle, upregulated INF-γ/TNF-α, and downregulated IL-10 in a co-culture system with T-cells.	MDA-MB-231
5	Wang, 2020 [13]	miR-570-3p	This miR can inhibit PD-L1 expression, stimulate apoptosis, and decrease tumor proliferation and migration via inhibiting the PI3K/Akt/mTOR signaling pathway.	MDA-MB-231, and MDA-MB-468
6	Peng, 2020 [33]	miR-200c-3p	The miR-200c-3p inhibitor can stimulate PD-L1 expression.	MDA-MB-231, and BT549
7	Dastmalchi, 2020 [34]	miR-424-5p	This miR can stimulate autophagy and apoptosis in tumoral cells. Its overexpression can decrease tumor proliferation via the suppressing of the PI3K/Akt pathway, arresting the cell cycle, inhibiting colony-formation of tumor cells, and repressing angiogenesis. With the inhibitory effect on tumoral PD-L1, this miR can upregulate INF-γ, TNF-α, and IL-2 and downregulate IL-10 expression in co-cultured T-cells.	MDA-MB-231
8	Azarbarzin, 2021 [35]	miR-383-5p	The restoration of this miR can inhibit PD-L1 expression, tumor invasion, clonogenicity, and proliferation of tumoral cells. Furthermore, this miR can arrest the cell cycle and stimulate apoptosis. In a co-culture system with T-cells, this miR can upregulate the expression of pro-inflammatory cytokines and downregulate the anti-inflammatory cytokines. The anti-tumoral effects of this miR might be attributable to its inhibitory effect on the PI3k/Akt signaling pathway.	MDA-MB-231
9	Zhao, 2019 [36]	miR-34a-5p	This miR can suppress tumoral PD-L1 expression, leading to the inhibition of tumor proliferation and migration both in vitro and in vivo.	MDA-MB-231
10	Li, 2019 [37]	miR-3609	This miR can inhibit the tumoral PD-L1 and improve the survival of mice bearing TNBC.	MDA-MB-231, and MDA-MB-468
11	Yang, 2018 [38]	miR-195-5p	This miR can downregulate tumoral PD-L1 expression in TNBC.	MDA-MB-231
12	Yang, 2018 [38]	miR-497-5p	This miR can downregulate tumoral PD-L1 expression in TNBC.	MDA-MB-231
13	Huang, 2017 [6]	miR-34a	This miR can inhibit tumoral PD-L1 and decrease the viability and invasion of tumor cells. In animal models, miR-34a restoration can inhibit tumoral PD-L1, increase CD8^+^ and CD4^+^ T-cells, and inhibit the recruitment of macrophages and Tregs into the tumor microenvironment.	MDA-MB-231, and HCC38

Abbreviations: miR: microRNA, PD-L1: programmed death-ligand 1, CD: cluster of differentiation, PI3K: phosphatidylinositol 3-kinase, Akt: protein kinase B, mTOR: mammalian target of rapamycin, EMT: epithelial-to-mesenchymal transition, INF-γ: interferon-γ, TNF-α: tumor necrosis factor-α, IL: interleukin, TNBC: triple-negative breast cancer, Treg: regulatory T-cell, and MMP: matrix metalloproteinase.

**Table 3 genes-12-01206-t003:** Evaluating the potential bias among in vitro studies.

No.	First Author, Publication Year	Was the Studied Cancer Cell Line(s) Reported?	Was the Duration of Exposure to the Studied miR to Tumoral Cells Reported?	Was the ConCentration of the Studied miR Reported?	Was a Standard Culture Media Used for the Study?	Were Reliable Tools Used to Assess the Outcome?	Were the Experiments Conducted More than Once?	Was More than One Independent Experiment Performed?	The Overall Risk of Bias
1	Narges Dastmalchi, 2021 [30]	Without bias	Without bias	Without bias	Without bias	Without bias	Without bias	Without bias	Without bias
2	Katayoun Bahman Soufiani, 2021 [31]	Without bias	Without bias	With bias	Without bias	Without bias	Without bias	Without bias	Low-bias
3	Mina Rasoolnezhad, 2021 [32]	Without bias	Without bias	Without bias	Without bias	Without bias	Without bias	Without bias	Without bias
4	Li-Li Wang, 2020 [13]	Without bias	Without bias	With bias	Without bias	Without bias	Without bias	Without bias	Low-bias
5	Fu Peng, 2020 [33]	Without bias	With bias	With bias	Without bias	Without bias	Without bias	Without bias	Low-bias
6	Narges Dastmalchi, 2020 [34]	Without bias	Without bias	Without bias	Without bias	Without bias	Without bias	Without bias	Without bias
7	Shirin Azarbarzin, 2021 [35]	Without bias	Without bias	Without bias	Without bias	Without bias	Without bias	Without bias	Without bias
8	Lianzhou Yang, 2018 [38]	Without bias	Without bias	Without bias	Without bias	Without bias	Without bias	Without bias	Without bias
9	Xiaojia Huang, 2017 [6]	Without bias	Without bias	With bias	Without bias	Without bias	Without bias	Without bias	Low-bias

**Table 4 genes-12-01206-t004:** Evaluating the potential bias among in vivo studies.

No.	First Author and Publication Year	Sequence Generation	Baseline Characteristics	Allocation Concealment	Random Housing	Blinding (Performance Bias)	Random Outcome Assessment	Blinding (Detection Bias)	Incomplete Outcome Data	Selective Outcome Reporting	Other Sources of Bias
1	Yueyuan Zhou, 2021 [12]	No bias	No bias	No bias	With Bias	No bias	No bias	With Bias	No bias	No bias	No bias
2	Qiuyang Zhao, 2019 [36]	No bias	No bias	No bias	With Bias	No bias	No bias	With Bias	No bias	No bias	No bias
3	Duolu Li, 2019 [37]	No bias	No bias	No bias	No bias	No bias	No bias	With Bias	No bias	No bias	No bias

**Table 5 genes-12-01206-t005:** The recent trend in the clinical trials for targeting PD-L1 for TNBC patients.

No.	Intervention	Mechanism of Action	Phase	Study Start Date	The Status	Clinicaltrials.gov Identifier
1	Atezolizumab and Paclitaxel	PD-L1 blockade and disrupting mitosis	III	25 Aug 2017	Active, not recruiting	NCT03125902
2	Atezolizumab and nab-Paclitaxel	PD-L1 blockade and disrupting mitosis	III	17 Dec 2019	Recruiting	NCT04148911
3	Atezolizumab and nab-Paclitaxel	PD-L1 blockade and disrupting mitosis	III	23 Jun 2015	Active, not recruiting	NCT02425891
4	Atezolizumab, radiation, and Talazoparib	PD-L1 blockade and inducing DNA damage	II	1 Apr 2021	Recruiting	NCT04690855
5	Atezolizumab, Paclitaxel, Doxorubicin/Epirubicin, and Cyclophosphamide	PD-L1 blockade, disrupting mitosis, DNA synthesis inhibition, and protein synthesis inhibition	III	2 Aug 2018	Recruiting	NCT03498716
6	Avelumab	PD-L1 blockade	III	Jun 2016	Active, not recruiting	NCT02926196
7	Atezolizumab	PD-L1 blockade	III	19 Dec 2017	Recruiting	NCT03281954
8	Atezolizumab, Pegylated liposomal doxorubicin, and Cyclophosphamide	PD-L1 blockade, DNA synthesis inhibition, and protein synthesis inhibition	II	1 Jun 2017	Recruiting	NCT03164993
9	Durvalumab and Olaparib	PD-L1 blockade and DNA repair inhibition	II	4 Oct 2018	Active, not recruiting	NCT03167619
10	Avelumab and Palbociclib	PD-L1 blockade and inhibiting DNA replication	I	11 Aug 2020	Recruiting	NCT04360941
11	Atezolizumab, Bevacizumab, Gemcitabine, and Carboplatin	PD-L1 blockade, inhibiting angiogenesis, DNA synthesis inhibition, and suppressing DNA synthesis	II	Feb 2021	Not yet recruiting	NCT04739670
12	Durvalumab and Carboplatin	PD-L1 blockade and DNA synthesis inhibition	II	29 Aug 2017	Active, not recruiting	NCT03206203
13	Durvalumab, Oleclumab, Paclitaxel, and Carboplatin	PD-L1 blockade, CD73 blockade, disrupting mitosis, and DNA synthesis inhibition	I/II	28 Dec 2018	Recruiting	NCT03616886
14	Atezolizumab and nab-Paclitaxel	PD-L1 blockade and disrupting mitosis	II	4 Feb 2016	Active, not recruiting	NCT02530489
15	Durvalumab and CFI-400945	PD-L1 blockade and PLK4 inhibition	II	19 Dec 2019	Recruiting	NCT04176848
16	Durvalumab	PD-L1 blockade	I/II	Nov 2015	Active, not recruiting	NCT02489448
17	Atezolizumab and Capecitabine	PD-L1 blockade and inhibiting DNA synthesis	II	15 Jan 2019	Recruiting	NCT03756298
18	Atezolizumab, Ipatasertib, and Paclitaxel	PD-L1 blockade, Akt inhibition and disrupting mitosis	III	25 Nov 2019	Active, not recruiting	NCT04177108
19	Atezolizumab, Bevacizumab, and Paclitaxel	PD-L1 blockade, inhibiting angiogenesis and disrupting mitosis	II	5 Oct 2020	Recruiting	NCT04408118
20	Atezolizumab, Gemcitabine, Capecitabine, and Carboplatin	PD-L1 blockade, DNA synthesis inhibition, inhibiting DNA synthesis and suppressing DNA synthesis	III	11 Jan 2018	Recruiting	NCT03371017

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
