# Peer review of "A Systematic Review on the Therapeutic Potentiality of PD-L1-Inhibiting MicroRNAs for Triple-Negative Breast Cancer: Toward Single-Cell Sequencing-Guided Biomimetic Delivery"

_genes, 2021, doi:10.3390/genes12081206_

Round 1
Reviewer 1 Report
The manuscript entitles “The current evidence on the therapeutic potentiality of microRNAs-targeting PD-L1 in triple-negative breast cancer: a systematic review” by Shadbad et al. is an interesting systematic review that included 11 studies published before 22 January 2021 and highlighted miR-424-5p, miR-676-3p, miR-570-3p, miR-200c, miR-383-5p, miR-34a, miR-3609, miR-195, and miR-497 that can downregulate the tumoral PD-L1 and inhibit the proliferation and migration of TNBC cells.
The paper is about an important subject, and it showed exciting results; however, some critical points and/or corrections should be performed to paper's acceptance, as reported below:
- There are formatting problems in all the text, such as lack of spaces (which seriously compromise the understanding of some ideas), paragraphs with erroneously bold and different font sizes.
Although formatting problems may seem like a minor problem, understanding has been impaired at various times. Here are some examples: “ICIscan”, “evidenceof”, “findingsin”, “miRsare”, “miRscan”.
- Important English review in all manuscript. For example, in the abstract what did the authors mean with “the current ICIscan not impede the de novo expression of PD-L1 on tumoral cells”?
- In “the introduction section” is important better describe the immunohistochemical subtypes to emphasize the importance of TNBC studies.
- Is essential to improve review about PD-1/PD-L1 in TNBC. There are more de 250 papers about the subject and, in the manuscript, they only cited two. Maybe, a good strategy is also include a recent review about the subject.
- Include more information about clinical trials of immunotherapy in TNBC.
- Many cited microRNAs influence the PI3K/Akt pathway. It is important to describe the relation of PI3K/Akt pathway and PD-L1 regulation in the manuscript.
- In miR-424-5p topic, what type of “bioinformatic results” has demonstrated that miR-424-5p expression is associated with overall survival? Is it analysis in public data as TCGA? Better explain these aspects.
- It’s important to describe the name e main function of many genes cited in the manuscript such as COPS5, NOL4L, LDHA, MUC1, etc.
- In miR-676-3p section, the idea about chemosensitivity of TNBC cells are not connected with the text. Better explain this idea.
- Define “MDSC” in the manuscript.
Author Response
The manuscript entitles "The current evidence on the therapeutic potentiality of microRNAs-targeting PD-L1 in triple-negative breast cancer: a systematic review" by Shadbad et al. is an interesting systematic review that included 11 studies published before 22 January 2021 and highlighted miR-424-5p, miR-676-3p, miR-570-3p, miR-200c, miR-383-5p, miR-34a, miR-3609, miR-195, and miR-497 that can downregulate the tumoral PD-L1 and inhibit the proliferation and migration of TNBC cells.
The paper is about an important subject, and it showed exciting results; however, some critical points and/or corrections should be performed to paper's acceptance, as reported below:
We are sincerely appreciative of the reviewer for her/his important input. We believe the suggested changes have substantially improved the quality of this paper over the previous version and helped us enrich our systematic review content. We have addressed all of the comments and revised the manuscript accordingly.
Query 1. here are formatting problems in all the text, such as lack of spaces (which seriously compromise the understanding of some ideas), paragraphs with erroneously bold and different font sizes. Although formatting problems may seem like a minor problem, understanding has been impaired at various times. Here are some examples: "ICIscan", "evidenceof", "findingsin", "miRsare", "miRscan".
We are thankful for the valuable comment of the reviewer. The reviewer has raised a valid concern. In this regard, we have thoroughly revised the manuscript and provided the readership with a neat and articulated manuscript.
Query 2. Important English review in all manuscript. For example, in the abstract what did the authors mean with "the current ICIscan not impede the de novo expression of PD-L1 on tumoral cells"?
We are appreciative of this valuable comment. The reviewer has raised a valid concern; thus, the manuscript was carefully revised.
Query 3. In "the introduction section" is important better describe the immunohistochemical subtypes to emphasize the importance of TNBC studies.
We are grateful for this valuable comment. According to the reviewer's comment, we explained more about the subtypes of breast cancers and highlighted the importance of TNBC.
- Response to the reviewer:
Breast cancer is among the frequently diagnosed malignancy among females. Yersal et al. have classified breast cancers into 5 groups, i.e., luminal-A, luminal-B, human epidermal growth factor receptor 2 (HER2)-positive, basal-like breast, and normal breast-like. Luminal-A is characterized by low proliferation-related genes expression and a high level of estrogen receptor (ER) expression and carries a relatively good prognosis. In contrast to luminal-A, luminal-B is characterized by increased expression of proliferation-related genes and carries a relatively poor prognosis. HER2-positive breast cancers are aggressive tumors characterized by increased HER2 expression and other genes related to this signaling pathway. Basal-like breast cancer is characterized by aggressive nature, high histological grade, high mitotic index, increased expression of myoepithelial markers, and no expression of ER, progesterone receptor (PR), and HER2. Clinically, they are regarded as TNBC; however, they are not completely synonymous. The normal breast-like group is a poorly defined group that usually does not respond to neoadjuvant chemotherapy. Since they do not express ER, PR, HER2, they might be regarded as triple-negative. However, their genetic profile is not consistent with basal-like breast cancers [1]. TNBC, which accounts for approximately 15% of breast cancer cases, does not overexpress the conventional receptors of breast cancer, i.e., PR, HER2, and ER [2]. Besides the lack of targets for target therapy, the highly aggressive nature of TNBC and the poor prognosis of TNBC patients can justify the need to develop novel therapeutic approaches to treat TNBC patients [3].
Query 4. Is essential to improve review about PD-1/PD-L1 in TNBC. There are more de 250 papers about the subject and, in the manuscript, they only cited two. Maybe, a good strategy is also include a recent review about the subject.
We are appreciative of this valuable comment. The reviewer has raised a valid concern; thus, we included more studies to highlight the importance of the PD-1/PD-L1 axis in TNBC.
- Response to the reviewer:
Inhibitory immune checkpoints are implicated in maintaining the immunosuppressive tumor microenvironment. The PD-L1/PD-1 axis is one of the critical and well-established inhibitory immune checkpoint axes that is highly implicated in immune evasion of tumoral cells. This axis can be established between tumoral cells and immune cells [4]. A recent meta-analysis has shown a strong association between tumor-infiltrating T-cells and PD-L1 in TNBC patients [5]. Based on a study, PD-L1 is overexpressed on 85% of TNBC, and its overexpression is associated with inferior overall survival and disease-free survival in patients with TNBC [6]. Also, it has been reported that TNBC cells express PD-L1 more than other breast cancer subtypes [7]. It has been shown that PD-L1 silencing can substantially decrease tumor viability, suppress tumor clonogenicity, arrest the cell cycle, stimulate apoptosis, inhibit tumor migration, upregulate pro-inflammatory cytokines, and downregulate anti-inflammatory cytokine a co-culture system with T-cells and TNBC cells [8]. Therefore, tumoral PD-L1 can be a valuable target for target therapy of TNBC cancer.
Query 5. Include more information about clinical trials of immunotherapy in TNBC.
We are thankful for the valuable comment of the reviewer. The reviewer has raised a valid concern. Based on the reviewer's comment, we have provided the readership with the recent trend in immunotherapy of TNBC patients in one table.
Query 6. Many cited microRNAs influence the PI3K/Akt pathway. It is important to describe the relation of PI3K/Akt pathway and PD-L1 regulation in the manuscript.
We want to extend our appreciation for the meticulous consideration given to our manuscript. We understand the concern of the reviewer. Therefore, we provided the readership with a heatmap to demonstrate the relationship between the studied miRs and the PI3K/Akt signaling pathway and discussed that in the pertained section.
Query 7. In miR-424-5p topic, what type of "bioinformatic results" has demonstrated that miR-424-5p expression is associated with overall survival? Is it analysis in public data as TCGA? Better explain these aspects.
We are thankful for the valuable comment of the reviewer. The reviewer has raised a valid concern. Thus, we did the requested revision.
- Response to the reviewer:
Their bioinformatic results from the METABRIC have demonstrated that miR-424-5p expression can be associated with improved overall survival in breast cancer patients (HR = 0.77, 95% CI: 0.63 – 0.94, and P-value = 0.0091) [12].
Query 8. It's important to describe the name e main function of many genes cited in the manuscript such as COPS5, NOL4L, LDHA, MUC1, etc.
We are appreciative of this valuable comment. The reviewer has raised a valid concern; we described the name and the main function of the genes.
Query 9. In miR-676-3p section, the idea about chemosensitivity of TNBC cells are not connected with the text. Better explain this idea.
We are grateful for this comment. This concern was addressed during the revision of the manuscript.
Query 10. Define "MDSC" in the manuscript.
We are appreciative of this comment. This concern was addressed during the revision of the manuscript.
Reviewer 2 Report
Reviewing report:
In this systematic review, Shadbad et al. summarize the recent findings in identifying some miRNAs (miR-424-5p, miR-676-3p, mir-570-3p, miR-200c, miR-383-5p, miR-34a, miR-3609, miR-195 and miR-497) and their effect on tumor development and anti-tumoral immune responses in TNBC. As mentioned by the authors, miR-based gene therapy can be an appealing approach to inhibit the de novo expression of PD-L1 and can prevent immune-related adverse events in affected patients.
Globally, I find this study well written and well designed especially if we consider the flowchart of study selection process (Figure 1). However, we can regret the very low number of studies included in the current systemic review.
General comments
- My major concerns relate more to the absence of a figure summarizing the effect of selected miRNAs on the expression of PD-L1 and the inhibition of tumor development in TNBC. This figure could also point out target genes and signaling pathways regulated by these miRNAs in TNBC.
- The authors mentioned the effect of studied miRNA on TNBC cell or tumor microenvironment of TNBC. Do the authors know whether the effect of these miRNAs on PD-L1 cascade may also occur in other solid tumors? Could the authors discuss this point please in the review please?
- In the discussion part, authors described in particular the effects of miR-424-5p on PI3K/Akt pathway in MDA-MB-231 cells...and a bit further in the same paragraph they highlighted the role of miR-424-3p in prostate cancer. As many studies have reported that miRNAs may have distinct biological functions in pathology, especially in cancer, could the authors fix this point, please.
….
Specific comments
- In the methods part, authors wrote “The Embase, Scopus, …, searched to obtain the relevant records published before 22 January 2020” whereas authors mentioned the date of 22 January 2021 in the summary part. Could the authors correct it please?
- In the figure legend of the Figure 1, please replace “he flowchart of study selection process” with “The flowchart…”
- In the column of the Table 2 dealing with the effect of studied miR on TNBC cell, please replace for miR-383-5p “The anti-tumoral effects of this miR might be attributable to its inhibitory effect n the PI3k…” with “its inhibitory effect on the PI3k”…
Author Response
In this systematic review, Shadbad et al. summarize the recent findings in identifying some miRNAs (miR-424-5p, miR-676-3p, mir-570-3p, miR-200c, miR-383-5p, miR-34a, miR-3609, miR-195 and miR-497) and their effect on tumor development and anti-tumoral immune responses in TNBC. As mentioned by the authors, miR-based gene therapy can be an appealing approach to inhibit the de novo expression of PD-L1 and can prevent immune-related adverse events in affected patients.
Globally, I find this study well written and well designed especially if we consider the flowchart of study selection process (Figure 1). However, we can regret the very low number of studies included in the current systemic review.
We are sincerely appreciative of the reviewer for her/his important input. We believe the suggested changes have substantially improved the quality of this paper over the previous version and helped us enrich our systematic review content. We have addressed all of the comments and revised the manuscript accordingly.
Query 1. My major concerns relate more to the absence of a figure summarizing the effect of selected miRNAs on the expression of PD-L1 and the inhibition of tumor development in TNBC. This figure could also point out target genes and signaling pathways regulated by these miRNAs in TNBC.
We are thankful for the valuable comment of the reviewer. The reviewer has raised a valid concern. In this regard, we provided the readership with a heatmap figure to demonstrate the enrichment of these PD-L1-inhabiting miRs and correlate with experimental findings and an illustration depicting the proposed strategy and the overall effect of these PD-L1-inhibiting miRs in TNBC.
Query 2. The authors mentioned the effect of studied miRNA on TNBC cell or tumor microenvironment of TNBC. Do the authors know whether the effect of these miRNAs on PD-L1 cascade may also occur in other solid tumors? Could the authors discuss this point please in the review please?
We want to extend our appreciation for the meticulous consideration given to our manuscript. We understand the concern of the reviewer. For this purpose, we specifically mentioned and discussed this crucial point in the discussion.
- Response to the reviewer:
Although the current systematic review has highlighted the promising results of PD-L1-inhibiting miRs restoration in treating TNBC, some points are worthy of mentioning. First, the recent advances in bioinformatic technologies have provided enormous data to predict the interactions between genes and miRs; therefore, these data can predict that specific miRs can target specific oncoproteins, like PD-L1. However, solely relying on these data, without investigating the overall effect of that miR resto-ration on the cell viability, migration, chemosensitivity, clonogenicity, apoptosis, etc., might be misleading. For instance, Chen et al. have reported that miR-191-5p can substantially inhibit PD-L1 expression in RKO cells, and they have not studied the overall effect of its restoration in colorectal cancer cells [87]. However, it has been reported that miR-191-5p can increase tumorigenicity and migration of colorectal cancer cells [88,89].
The third consideration is related to safe, specific, and effective miR delivery because one tumor-suppressive miR in one cancer can promote tumorigenicity in other tissues. For instance, Mansoori et al. have demonstrated that miR-142-3p can stimulate apoptosis and arrest the cell cycle at the G2/M phase in breast cancer cells [92]. However, Zhu et al. have shown that miR-142-3p can substantially inhibit apoptosis and increase the cell viability of gastric cancer cells [93]. Therefore, there is a need to develop a specific, safe, and effective miR delivery approach.
Query 3. In the discussion part, authors described in particular the effects of miR-424-5p on PI3K/Akt pathway in MDA-MB-231 cells...and a bit further in the same paragraph they highlighted the role of miR-424-3p in prostate cancer. As many studies have reported that miRNAs may have distinct biological functions in pathology, especially in cancer, could the authors fix this point, please.
We are appreciative of this valuable comment. The reviewer has raised a valid concern; thus, we have removed that sentence and specifically mentioned and discussed this crucial point in the discussion to address this critical issue.
- Response to the reviewer:
The second consideration should be devoted to the importance of the biological differences of -3p and -5p miRs. For instance, Yang et al. have shown that inhibiting miR-193a-5p and docetaxel treatment has remarkably decreased prostate cancer volume compared to treatment with docetaxel and control miR in mice models [90]. However, Liu et al. have demonstrated that miR-193a-3p transfection can decrease tumor proliferation, arrest the cell cycle, inhibit cell viability, and suppress clonogenicity of prostate cancer cells [91].
Query 4. In the methods part, authors wrote "The Embase, Scopus, …, searched to obtain the relevant records published before 22 January 2020" whereas authors mentioned the date of 22 January 2021 in the summary part. Could the authors correct it please?
We are grateful for this comment. The reviewer has raised a valid concern; thus, we corrected the search date.
- Response to the reviewer:
The Cochrane Library, Embase, Scopus, PubMed, ProQuest, Web of Science, Ovid, and IranDoc were systematically searched to obtain peer-reviewed records published before 25 May 2021 without any restrictions in language or country.
Query 5. In the figure legend of the Figure 1, please replace "he flowchart of study selection process" with "The flowchart…"
We are thankful for this comment. The reviewer has raised a valid concern; thus, we corrected the aforementioned statement.
- Response to the reviewer:
Figure 1. The flowchart of the study selection process.
Query 6. In the column of the Table 2 dealing with the effect of studied miR on TNBC cell, please replace for miR-383-5p "The anti-tumoral effects of this miR might be attributable to its inhibitory effect n the PI3k…" with "its inhibitory effect on the PI3k"…
We are grateful for this comment. The reviewer has raised a valid concern; thus, we corrected it.
- Response to the reviewer:
The anti-tumoral effects of this miR might be attributable to its inhibitory effect on the PI3k/Akt signaling pathway.
Reviewer 3 Report
This is a well-written systematic review, pointing out the importance of certain miRNAs for the tumoral PD-L1 expression in TNBC. More than 1 independent reviewers participate in the selection of each phase.
Some suggestions below:
- The article used 4 electronic databases. Multifaceted search including database (google scholar), Trial registries, Conference proceedings, and the thesis will make the search for relevant studies more exhaustive.
- A paragraph is required to specify the methodological quality of included studies and how to assess them.
- It will be better to specify the relative expression level of these certain miRNAs in TNBC cells comparing with the normal cells.
- Briefly discuss the potential strategies for the miRNA-based therapy for TNBC patients.
Author Response
This is a well-written systematic review, pointing out the importance of certain miRNAs for the tumoral PD-L1 expression in TNBC. More than 1 independent reviewers participate in the selection of each phase.
We are sincerely appreciative of the reviewer for her/his important input. We believe the suggested changes have substantially improved the quality of this paper over the previous version and helped us enrich our systematic review content. We have addressed all of the comments and revised the manuscript accordingly.
Query 1. The article used 4 electronic databases. Multifaceted search including database (google scholar), Trial registries, Conference proceedings, and the thesis will make the search for relevant studies more exhaustive.
We are thankful for the valuable comment of the reviewer. The reviewer has raised a valid concern. Therefore, we included Ovid, IranDoc, ProQuest, and the Cochrane Library databases to make our search more exhaustive. Indeed, we included the databases that they allow to perform the advanced search with our 91 keywords.
- Response to the reviewer:
The Cochrane Library, Embase, Scopus, PubMed, ProQuest, Web of Science, Ovid, and IranDoc were systematically searched to obtain peer-reviewed records published before 25 May 2021 without any restrictions in language or country. For this sys-tematic search, we used the following keywords: (“BT20” OR “BT-549” OR “Cal51” OR “HCC 38” OR “HCC 1143” OR “HCC 1187” OR “HCC 1395” OR “HCC 1599” OR “HCC 1739” OR “HCC 1806” OR “HCC 1937” OR “HCC 2157” OR “HCC 2185” OR “HCC 3153” OR “HS578T” OR “MDA-MB-157” OR “MDA-MB-231” OR “MDA-MB-435” OR “MDA-MB-436” OR “MDA-MB-468” OR “SUM52PE” OR “SUM102PT” OR “SUM149PT” OR “Sum159PT” OR “SUM185PE” OR “SUM229PE” OR “SUM1315M02” OR “triple” OR “triple negative breast cancer” OR “tri-ple-negative” OR “triple negative” OR “triple-negative breast cancer” OR “TNBC” OR “HCC70” OR “DU4475” OR “Hs 578T” OR “MDA-MB-453” OR “HCC1599” OR “HCC1937” OR “HCC1143” OR “HCC38” OR “HCC1806” OR “HCC1187” OR “HCC1395” OR “CRL-2331” OR “CRL-2336” OR “CRL-2321” OR “HTB-132” OR “CRL-2314” OR “CRL-2315” OR “CRL-2335” OR “CRL-2322” OR “HTB-123” OR “HTB-122” OR “HTB-126” OR “HTB-26” OR “HTB-130” OR “HTB-24” OR “HTB-131” OR “HTB-19” OR “CRL-2324” OR “triple negative breast neoplasms”) and (“CD274” OR “programmed death-ligand 1” OR “PD-L1” OR “PDL1” OR “B7-H1” OR “PDCD1LG1” OR “B7 H1” OR “B7H1” OR “HPD-L1” OR “cluster of differentiation 274” OR “PD L1” OR “CD 274” OR “cluster of differentiation274” OR “B7 homolog 1” OR “PDCD1 ligand 1” OR “PDCD1L1” OR “B7-H1 antigen” OR “programmed death 1 ligand 1”) and (“miRNA-“ OR “microRNA-“ OR “microRNAs” OR “miR-“ OR “micro RNA” OR “microRNA” OR “micro RNA-“ OR “miRNA” OR “miR” OR “miRNA-“ OR “miRNA”). We extracted the names of TNBC cell lines from the study conducted by Chavez et al. [15]. We also incorporated Emtree and MeSH terms to increase the sensitivity of our systematic review.
Query 2. A paragraph is required to specify the methodological quality of included studies and how to assess them.
We are grateful for this valuable comment. The reviewer has raised a valid concern. Therefore, we applied two checklists to appraise the quality of the included studies.
- Response to the reviewer:
To increase the transparency of the obtained results and highlight the potential risk of bias among the included studies, two quality assessment tools were used. For in vivo studies that investigated the restoration of PD-L1-inhibiting miR in animal models, we used the SYRCLE's RoB checklist [16]. For in vitro studies, we used a previously applied checklist for assessing the potential risk of bias among in vitro studies [17].
Query 3. It will be better to specify the relative expression level of these certain miRNAs in TNBC cells comparing with the normal cells.
We are appreciative of this comment. We understand the concern of the reviewer; thus, we specified the relative expression level of the miRs in TNBC in the pertained sections of the discussion.
- Response to the reviewer:
- Wang et al. have demonstrated that miR-424-5p is substantially downregulated in TNBC tissues and MDA-MB-468 cells compared to adjacent normal tissues and MCF-10A.
- Zhao et al. have demonstrated that the expression of miR-138-5p is substantially downregulated in MDA-MB-231 cells compared to MCF-10A.
- Besides, the expression level of miR-570-3p is substantially decreased in TNBC tissues compared to adjacent normal tissues [47].
- Ren et al. have shown that miR-200c-3p expression is considerably downregulated in MDA-MB-231 cells and TNBC tissues compared to MCF-10A and adjacent normal tissues, respectively.
- It has been reported that the expression of miR-383-5p is substantially decreased in MDA-MB-231 cells compared to MCF-10A cells.
- Li et al. have shown that miR-34a-5p is substantially downregulated in MDA-MB-231cells compared to MCF-10A cells [55].
- Besides, the expression of miR-3609 is substantially decreased in MDA-MB-231 and MDA-MB-468 cells compared to HBL-100.
- Wang et al. have reported that miR-195-5p is substantially decreased in MDA-MB-435 and MDA-MB-231 cells compared to MCF-10A cells.
- It has been reported that miR-497-5p has been considerably downregulated in TNBC tissues, MDA-MB-453, MDA-MB-468, and MDA-MB-231 cells compared to non-tumoral tissues and MCF-10A cells [70,71].
Query 4. Briefly discuss the potential strategies for the miRNA-based therapy for TNBC patients.
We want to extend our appreciation for the meticulous consideration given to our manuscript. We totally understand the reviewer's concern; thus, we briefly discussed one of the promising methods for miR delivery and provided the readership with state-of-art findings in this field.
- Response to the reviewer:
4.2 The delivery of miRs for treating TNBC: toward effective, biocompatible, and safe delivery
miR-based gene therapy is an appealing approach for treating malignancies. Although miR-based gene therapy has not been investigated in phase III clinical trials, it is a promising approach to modulate the expression of oncoproteins [74]. However, the safety, specificity, and efficacy of their delivery might be considered a daunting challenge. The ideal vehicle for miR delivery should specifically deliver cargo to tumoral cells and not induce toxicity. In this regard, polymer-based vectors and biomimetic carriers have been investigated in preclinical settings [75]. Herein, we aim to discuss the recent advances in miR-delivery via biomimetic carriers briefly. Besides, we intend to introduce recent advancements in single-cell sequencing and highlight the potentiality of combining this technology with biomimetic carriers to provide the affected patients with personalized, specific, and safe tumor-suppressive miR delivery.
4.2.1 Biomimetic carriers
The fabrication of membranes of red blood cells, white blood cells, platelets, and tumoral cells can decrease the immunogenicity and clearance of carriers. Besides, they can provide the carriers with targeted release and enhance their biocompatibility. In this regard, the biomimetic carriers derived from the leukocytes membrane have demonstrated promising results because of their inherited ligands. During inflammation, the interaction between the endothelial cells and circulating leukocytes paves the way for the infiltration of leukocytes into the inflamed site [76]. Therefore, coating the leukocyte membrane into carriers can provide us with the targeted release. Besides, the biocompatible nature of coated membranes can improve the biocompatibility of carriers and decrease toxicity and fast clearance from circulation. Since discussing all types of biomimetic carriers is out of the scope of the current study, we briefly discuss the recent advances in this field for cancer therapy.
Leukosomes are a recently introduced class of biomimetic nanovesicles that contain the specific molecules for cellular adhesion, e.g., P-selectin glycoprotein ligand-1 and lymphocyte function-associated antigen 1, among others. Recently, Molinaro et al. have developed leukosomes that can preserve the pharmaceutical properties of liposomes, and they predominantly accumulate in the tumor microenvironment in mice bearing TNBCs [77]. Martinez et al. have shown that the tumor microenvironment accumulation ability of leukosomes compared to liposomes is 16-fold. Also, leukosomes, compared to liposomes, have been predominately present in the tumoral vessels wall and lumen. However, the blockade of lymphocyte function-associated antigen 1 has substantially (around 95%) decreased the ability of leukosomes to accumulate in the tumor [78].
Neutrophils are the most abundant leukocytes in the human. Since activated neutrophils can target circulating tumor cells and tumor microenvironment, neutrophil membrane-coated carriers have been investigated for targeted delivery. Wang et al. have demonstrated that neutrophil membrane-coated carriers can escape from macrophage-mediated clearance and accumulate in the inflamed site [79]. Kang et al. have demonstrated that coating neutrophil membrane to PLGA-based nanoparticles can remarkably capture circulating tumor cells both in vitro and in vivo. Besides, these neutrophil membrane-coated carriers have been mainly accumulated in the premetastatic niche of mice bearing TNBC cells [80]. Cao et al. have reported that neutrophil membrane-coated polyethylene glycol (PEG)-PLGA-based carriers selectively accumulate in the tumor microenvironment following systematic administration. Besides, loading cytotoxic agents to this complex can substantially prolong the survival of animal models, decrease tumor size, and inhibit tumor metastasis [81]. Despite the promising results of neutrophil membrane-coated carriers in inhibiting tumor development, the inflamed sites other than the tumor microenvironment might facilitate their accumulation and decrease their specificity. Therefore, further investigations are needed to address this issue.
Unlike innate immune cells, T-cells can specifically recognize antigens. For developing T-cell-mediated immune responses, antigen-presenting cells should present antigens to naïve T-cells. Afterwards, naïve T-cells differentiate into CD4+ or CD8+ T-cells. The vast repertoire of T-cells can provide ample opportunity to develop specific T-cells for each antigen. Therefore, T-cell-membrane-coated carriers can pave the way for introducing specific carriers for miR delivery [82]. Although recent advances in single-cell sequencing have allowed us to investigate and profile tumoral cells at single-cell levels (see below), the high specificity of T-cell receptors might be an obstacle to address the vast inter-and intra-heterogeneity of tumor bulk. In this regard, metabolic glycoengineering can introduce different chemical groups to cellular glycans, which can serve as promising ligands for targeted therapy. Han et al. have intratumorally administrated bicyclo [6.1.0] nonyne (BCN) modified unnatural sugars (Ac4ManN-BCN) to produce ligands for their T-cell-membrane-coated carriers. Their azide-based T-cell-membrane-coated carriers have substantially accumulated in the tumor microenvironment [83]. Recently, Kang et al. have proposed another strategy to facilitate anti-tumoral immune responses via the application of T-cell-membrane-coated carriers. They have developed PLGA-based T-cell-membrane-coated carriers to stimulate anti-tumoral immune responses and deliver cytotoxic agents. Their proposed mechanisms for tumor rejection have been based on five items. First, these nanoparticles can be actively accumulated in the tumor microenvironment due to the presence of inherited lymphocyte function-associated antigen 1 on the nanoparticles and intercellular adhesion molecule-1 on the affected endothelial cells in the tumor tissue. Second, their infiltration is not influenced by the highly immunosuppressive tumor microenvironment. Third, the inherited Fas ligand of nanoparticles can activate apoptosis in tumoral cells. Fourth, they can occupy inhibitory immune checkpoint molecules and liberate the anti-tumoral immune cells from the immune checkpoint-mediated exhaustion. Besides, they can be loaded with cytotoxic factors to attenuate tumoral proliferation. Indeed, their in vivo results have indicated that these biomimetic carriers can remarkably prolong the survival of animal models [84]. Another approach to specifically target tumoral cells is developing CAR-T-cell-membrane-coated carriers. Recently, Ma et al. have extracted the membrane of CAR-T-cells that specifically target glypican-3, which is highly expressed in the studied tumor cells. Their results have indicated that delivery via CAR-T-cell-membrane-coated carriers can remarkably decrease the tumor weight compared to delivery via T-cell-membrane-coated carriers and mesoporous silica nanoparticles in animal models [85].
4.2.2 Single-cell sequencing technologies and biomimetic carriers: Future perspectives
Single-cell sequencing has provided ample opportunities to study tumoral cells at the single-cell level. The enormous data provided by single-cell sequencing technologies have furthered our knowledge of the vast inter-and intra-heterogeneity and provided answers for the failure of conventional anti-cancer therapies. Indeed, tumor bulk is not considered cells that have unified gene mutations rather the subpopulations of tumor bulk with distinct genetic properties [86]. The heterogeneity among the tumoral cells might be a daunting challenge for identifying suitable targets for active-targeting delivery. However, single-cell sequencing can allow us to identify personalized neo-antigens for each affected patient. Ideally, these neo-antigens should be expressed in tumor cells, and the healthy cells should minimally (or ideally not) express these antigens [40]. Therefore, fabricating the pertained CAR or T-cell receptor (TCR) or pep-tides to the above-mentioned biocompatible carriers can ameliorate the delivery of the PD-L1-inhibiting miRs and substantially inhibit TNBC development (Fig. 4). Although the recent study by Ma et al. has demonstrated promising results about the efficacy of specific CAR-T-cell-membrane-coated carriers [85], further studies are needed for the translation of this novel approach into clinical practice.